# Measuring the orbital angular momentum spectrum of an electron beam

Vincenzo Grillo[1,2], Amir H. Tavabi[3], Federico Venturi[1,4], Hugo Larocque[5], Roberto Balboni[6], Gian Carlo Gazzadi[1], Stefano Frabboni[1,4], Peng-Han Lu[3], Erfan Mafakheri[4], Frédéric Bouchard[5], Rafal E. Dunin-Borkowski[3], Robert W. Boyd[5,7], Martin P.J. Lavery[8], Miles J. Padgett[8] & Ebrahim Karimi[5,9]

Electron waves that carry orbital angular momentum (OAM) are characterized by a quantized and unbounded magnetic dipole moment parallel to their propagation direction. When interacting with magnetic materials, the wavefunctions of such electrons are inherently modified. Such variations therefore motivate the need to analyse electron wavefunctions, especially their wavefronts, to obtain information regarding the material's structure. Here, we propose, design and demonstrate the performance of a device based on nanoscale holograms for measuring an electron's OAM components by spatially separating them. We sort pure and superposed OAM states of electrons with OAM values of between −10 and 10. We employ the device to analyse the OAM spectrum of electrons that have been affected by a micron-scale magnetic dipole, thus establishing that our sorter can be an instrument for nanoscale magnetic spectroscopy.

[1] CNR-Istituto Nanoscienze, Centro S3, Via G Campi 213/a, I-41125 Modena, Italy. [2] CNR-IMEM Parco Area delle Scienze 37/A, I-43124 Parma, Italy. [3] Ernst Ruska-Centre for Microscopy and Spectroscopy with Electrons, Forschungszentrum Jülich, 52425 Jülich, Germany. [4] Dipartimento FIM Universitá di Modenae Reggio Emilia, Via G Campi 213/a, I-41125 Modena, Italy. [5] Department of Physics, University of Ottawa, 25 Templeton Street, Ottawa, Ontario, Canada K1N 6N5. [6] CNR-IMM Bologna, Via P. Gobetti 101, 40129 Bologna, Italy. [7] Institute of Optics, University of Rochester, Rochester, New York 14627, USA. [8] School of Physics and Astronomy, Glasgow University, Glasgow, G12 8QQ Scotland, UK. [9] Department of Physics, Institute for Advanced Studies in Basic Sciences, 45137-66731 Zanjan, Iran. Correspondence and requests for materials should be addressed to V.G. (email: vincenzo.grillo@unimore.it) or to E.K. (email: ekarimi@uottawa.ca).

Quantum complementarity states that particles, for example, electrons, can exhibit wave-like properties such as diffraction and interference upon propagation. Electron waves defined by a helical wavefront are referred to as twisted electrons and are imbued with several additional mechanical and magnetic properties[1]. For instance, upon elastic interaction, these magnetic properties, in conjunction with the opposite handedness of twisted electrons, allow for probing magnetic chirality as well as magnetic dichroism[2,3]. Furthermore, the added unbounded twisted motion of these charged particles is pertinent to the investigation of the nature of radiation[4], virtual forces and increasing the lifetime of unstable and metastable particles[5]. From a more fundamental viewpoint, structured electrons also provide novel insights into the quantum nature of electromagnetic–matter interaction, for example, the realization of Landau–Zeeman states[6,7] and spin-to-orbit coupling[8]. Orbital angular momentum (OAM)-carrying electron waves can be generated through a variety of methods by directly interacting with the electrons' wavefronts[9]. These processes rely on devices such as spiral phase plates[10], amplitude and phase holograms[11–14], cylindrical lens[15] mode converters or even electron microscope corrector lenses[16]. Spin-to-orbit coupling has also been theoretically proposed as a method to add OAM to electrons[17]. Other methods to do so exploit the magnetic properties of electrons, most notably by employing a magnetic needle simulating a magnetic monopole[18].

Reciprocally, devices that are used to generate twisted electrons can also be adapted to measure an electron's OAM content[19]. The most commonly employed of these devices relies on a series of projective phase flattening measurements, allowing one to obtain the magnitude of each OAM component of a beam's spectrum[20]. To perform such an analysis, an OAM component's wavefront is flattened, thus causing it to gain a Gaussian-like profile upon propagation[21]. This profile allows it to be easily selected from the remaining parts of the beam and therefore to evaluate the intensity of this component. To obtain these relative intensities, each component needs to be coupled to the device's electron detection mechanism. However, this coupling process is biased towards electrons carrying lower absolute values of OAM, thus introducing discrepancies in the measured spectrum[22]. Moreover, much like how different types of OAM-carrying beams are generated using different devices, different devices are required to flatten the wavefronts of the beam's various OAM components. Therefore, though such an analysis of a beam's OAM content is efficient, it also requires a substantial number of elements to perform repetitive measurements. There are also other OAM measurement methods relying on interferometry[23,24], though they possess serious limitations such as a limited amount of information that can be extracted from the obtained interference patterns or also by the need for an extremely stable interferometer. Interferometry is also of limited usefulness when analysing the OAM content of inelastically scattered electrons due to their short coherence lengths.

A viable alternative to these methods is available in optics and relies on transforming an OAM-carrying photon's azimuthal phase variations into transverse phase gradients that can be spatially resolved and separated with a lensing element[25,26]. The device thus effectively behaves as an OAM spectrometer, which could be of significant interest in electron optics and materials science as it would provide detailed information on a material's magnetic spectrum.

The operation of our presented electron OAM spectrometer relies on a similar OAM separation process. Much like its optical counterpart, it is essentially based on unwrapping the azimuthal phase variations associated with an electron's OAM into variations over a Cartesian coordinate of the plane transverse to the electron's propagation. This effectively causes the electron's original helical wavefronts to become planar and inclined with respect to the beam's original direction of propagation. Namely, the degree to which these wavefronts are tilted will increase with the azimuthal variation of the electron's original phase profile, that is, its OAM. As a result, focusing these unwrapped waves with a lensing element will cause electrons originally carrying different OAM values to focus at correspondingly separate lateral positions. By using this method, we are thus able to decompose an electron beam's OAM content by measuring the relative electron intensity at each of these possible focusing positions.

## Results

**Theory**. The unwrapping process, as detailed in ref. 25, requires the beam's transverse profile in polar coordinates $(r, \varphi)$ to be physically mapped to Cartesian coordinates. Such a transformation can be achieved by means of a conformal mapping between the Cartesian coordinates of the initial beam's profile $(x, y)$ and those of its final profile $(u, v)$. The coordinates $(u, v)$ are log-polar coordinates and can be related to the Cartesian coordinates $(x, y)$ via the transformations $x = \exp(u)\cos(v)$ and $y = \exp(u)\sin(v)$ or equivalently by $u = \ln\left(\sqrt{x^2 + y^2}\right)$ and $v = \arctan(y/x)$[27]. A Cartesian coordinate $x + iy$ in the complex plane can be mapped onto its corresponding log-polar coordinate $u + iv$ with the conformal mapping $f(z) = \ln(z)$. Using a scalable version of this mapping, $\Phi(z) = a \ln(z/b)$, an OAM-carrying beam's transverse wavefunction $\psi_\ell(r, \varphi) = f(r)\exp(i\ell\varphi)$ can be mapped to the following wavefunction

$$\Phi(f(r)\exp(i\ell\varphi)) = U + i V$$
$$= a\ln(f(r)/b) + i\,a\ell\varphi. \qquad (1)$$

Such mappings are commonly conducted using a set of confocal phase elements performing a log-polar coordinate transformation[27].

**Implementation**. We adopt a similar diffractive approach to develop our sorter using two electron phase holograms, as displayed in Fig. 1, and where the lensing effects required to perform the mode transformation are configured using our electron microscope's lenses (see Supplementary Note 1 and Supplementary Fig. 1 for more details). The first of the holograms effectively maps the electron's azimuthal phase variations along a Cartesian coordinate, while the second provides phase corrections to defects introduced by the first hologram. Additional information concerning these elements is provided in Supplementary Note 2. We depict this process in Fig. 1 where we show an electron beam's recorded transverse profile as it propagates through the sorter. Here, we use electrons consisting of a superposition of states defined by OAM values of $\pm 5$, that is, $\left(\psi_{+5}(r, \varphi) + \psi_{-5}(r, \varphi)\right)/\sqrt{2}$ (ref. 28). Such a beam consists of a series of 10 lobes that are equidistantly arranged along a ring-shaped outline in the beam's transverse plane. The presence of these lobes is a direct result of the beam consisting of a superposition of two OAM components defined by the same magnitude, yet by opposite signs. More specifically, as they arise from the beam's OAM content, these lobes are additionally related to its constituent $\ell = \pm 5$ electrons' transverse azimuthal phase profiles. The beam was generated using the phase mask whose transmission electron microscopy image along with an image of the beam transmitted through such a hologram can be found in the generator plane of the sorter found in Fig. 1. After passing through the sorter's first hologram, these lobes rearrange themselves into a line since the beam's azimuthal phase variations have been unwounded along one of the Cartesian coordinates of

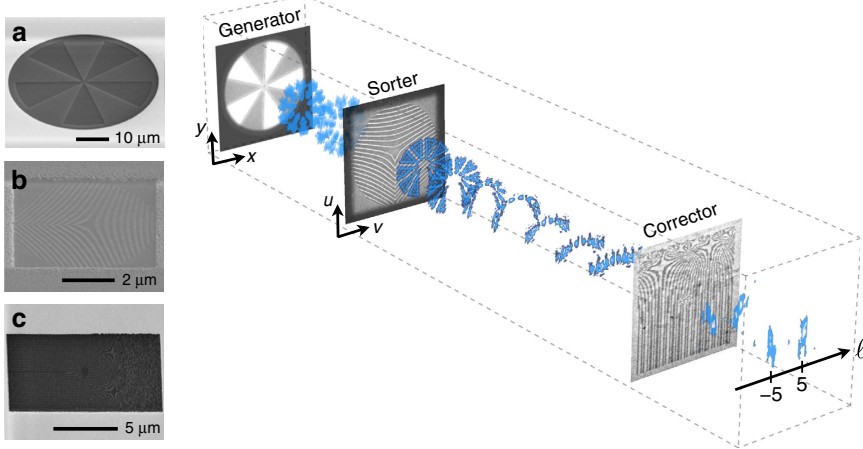

**Figure 1 | Schematics of the electron sorter.** These schematics also show an electron beam's experimental transverse intensity profile recorded at various planes in the sorting apparatus. A hologram in the sorter's generator plane that corresponds to the electron microscope's condensor, produces an electron beam carrying orbital angular momentum (OAM). In this particular case, the beam consists of a superposition of ±5 OAM states. The beam then goes through a hologram in the apparatus' sorter plane, positioned at the microscope's sample holder, that performs the required conformal mapping $(x, y) \mapsto (u, v)$. Once the beam is unwrapped, it passes through a hologram in the sorter's corrector plane corresponding to the microscope's selected area diffraction (SAD) aperture. This hologram brings corrections to any phase defects to the beam to stabilize its propagation through the rest of the sorter. At the sorter's output, the original beam's OAM content is spatially resolved on a screen and captured by a CCD camera. A more detailed schematic of our sorter's implementation is included in Supplementary Fig. 1 and Supplementary Note 1 where we provide details concerning the electron microscope lenses required to perform the mapping. Scanning electron microscopy (SEM) images of the depicted holograms, the ones in the generator, sorter and corrector planes, are shown in **a–c**, respectively.

its transverse plane. The beam propagates through the second hologram, shown in the corrector plane of Fig. 1, to stabilize its propagation. The beam then propagates through a lens and is focused onto a screen. This allows for the initial beam's OAM content to be readily analysed as depicted in the final plane of Fig. 1.

A more thorough calibration of the sorting apparatus was then performed by repeating the above process for wavefunctions carrying various values of OAM generated with various devices that are further discussed in Supplementary Note 3. The spectra resulting from this calibration have been background subtracted and deconvoluted using conventional spectroscopy methods (see Supplementary Note 4 and Supplementary Fig. 2). These results are displayed in Fig. 2 for the case of wavefunctions defined by single and superposed OAM states ranging from −10 to 10. Based on the various parameters defining the phase profile of the holograms used in the sorter, the cross-talk between components of the electrons' OAM spectra is expected to be below 20%. However, due to fabrication and alignment imperfections in our apparatus, including the holograms generating the OAM-carrying electrons, we observe higher values of cross-talk between OAM components. The values of the cross-talk observed in the spectra shown in Fig. 2 were found to be 28% (Fig. 2a), 43% (Fig. 2b), 39% (Fig. 2c) and 18% (Fig. 2d). Such experimental limitations could be overcome by adopting a fan-out configuration as employed in optical sorters[25,26] and by improving the quality of the sorter and the corrector holograms via alternative fabrication methods. Cross-talk can also be further reduced by an improved control over the electron beam's aberrations. Though this may be the case, the sorter's performance is clearly observed through the distinct separation of the OAM states contained in the specific electron beams.

This sorter can be used to analyse magnetic structures affecting the OAM content of an electron beam. Here, we use our sorter to analyse the magnetic properties of a magnetized sample deposited using the method described in ref. 29, a cobalt magnetic dipole in our case. To do so, the dipole has been positioned in the sorting apparatus in a way to replace the holograms that were previously

generating the test wavefunctions. This magnetic structure consists of a single magnetic bar. A scanning electron microscopy image of the bar can be found in Fig. 3a. Its magnetic configuration is characterized by a strong elongation enforcing the presence of a strong net magnetic dipole moment even after the magnetizing field is removed. Its remanence field was also characterized using electron holography and Lorentz imaging and is depicted in Fig. 3b.

The potential use of the sorter for this particular measurement arises from the fact that the multiplicative term introduced by a magnetic dipole onto a passing Gaussian electron beam wavefunction can be written as

$$g(r, \varphi) = \exp(i \chi(r) \sin \varphi) \qquad (2)$$

where $\chi(r) = (e\mu_0 \mathcal{M})/(hr)$, $h$ is the Planck constant, $\mu_0$ is the permeability of vacuum and $\mathcal{M}$ is the dipole's magnetic moment (see Supplementary Note 5 for more details). To observe the effect that such a term will have on an electron's OAM content, $g(r, \varphi)$ must be Fourier-expanded in terms of $\varphi$, namely $g(r, \varphi) = \sum_{\ell=-\infty}^{\infty} c_\ell(r) \exp(i\ell\varphi)$, where $c_\ell(r)$ are expansion coefficients. However, given that the resulting expansion terms carry a quantized azimuthal phase defined by $\ell$, then it follows that these components also carry OAM values of $\ell$. By default, the expansion coefficients $c_\ell$ correspond to the weight of each OAM component of a beam that has been affected by the magnetic dipole. These coefficients can be found using the Jacobi–Anger expression, that is, $\exp(i \chi \sin \varphi) = \sum_{\ell=-\infty}^{\infty} J_\ell(\chi) \exp(i\ell\varphi)$, and thus correspond to $J_\ell(\chi(r))$, where $J_\ell$ is the $\ell^{\text{th}}$ order Bessel function of the first kind. This formulation of the added phase clearly depicts the imparted OAM content of the dipole onto the electron beam's wavefunction and hence the ability to use the sorter to measure this magnetic dipole moment $\mathcal{M}$.

Though a more detailed analysis can be used to predict the OAM spectrum induced by the magnetic dipole, one can instead employ a simplified model to do so based on the apparatus' layout. The use of this model is justified by the truncation of the beam after passing through the first of the sorter's holograms. Such truncations result in lensing effects that will consequently displace

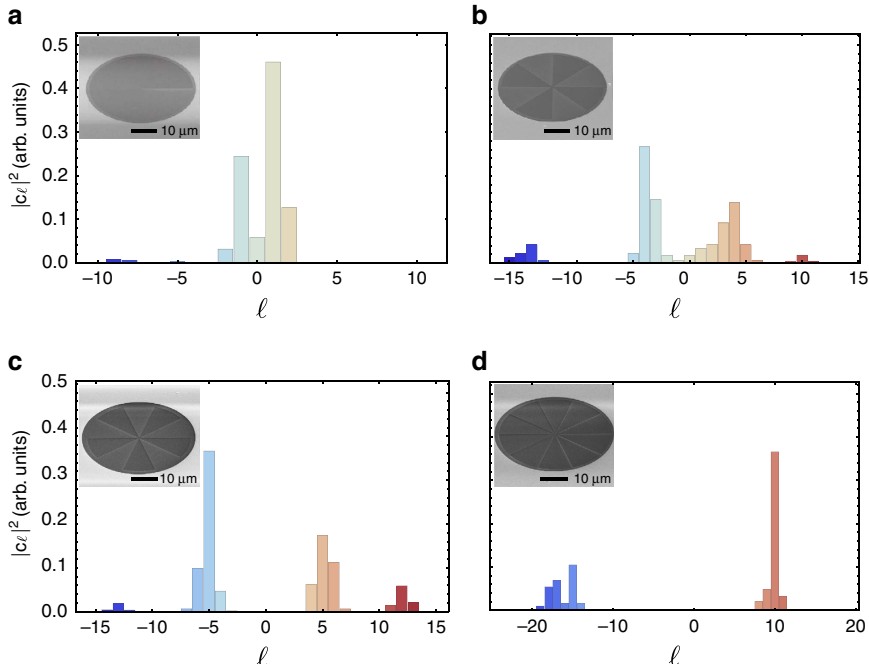

**Figure 2 | Experimental OAM spectra of electron beams.** Spectrum of a beam consisting of electrons defined by: (**a**) OAM of $+1$, $\psi_{+1}$, produced with a spiral phase plate; (**b**) a superposition of $\pm 4$ OAM states, $(\psi_{+4} + \psi_{-4})/\sqrt{2}$, generated by a phase mask; (**c**) a superposition of $\pm 5$ OAM states, $(\psi_{+5} + \psi_{-5})/\sqrt{2}$, generated by a phase mask; and (**d**) OAM of $+10$, $\psi_{+10}$, produced by a spiral phase plate. Scanning electron microscopy (SEM) images of the devices used to generate the analysed electron beams are provided in the insets of their respective spectra.

electrons that were originally positioned at the hologram's cutoff radius $r_{max}$. As a result, these electrons will occupy a greater area of the beam following the truncation. Their profile will therefore also be spatially extended in the observed OAM spectrum. This will cause the general outline of the sorter's output to be predominantly associated with these electrons. Strictly speaking, the output spectrum of the sorter will be determined by its numerical aperture. Therefore, we may approximate the phase added by the dipole onto the beam as $\chi(r)\sin\varphi \approx \chi(r_{max})\sin\varphi$ that effectively removes the need to consider the radial dependence of the beam's OAM. From this analysis, the relative probability of finding an electron of OAM $\ell\hbar$ in the resulting beam is given by the coefficient $|c_\ell|^2 = |J_{|\ell|}(\chi(r_{max}))|^2$ that will also yield the beam's observed OAM distribution as a function of $\ell$. This analysis, in agreement with a more rigorous approach based on numerical calculations and also an analytical approach (see Supplementary Note 5), reveals that the dominant decomposition coefficients of such an electron's spectrum will be attributed to $\ell$ values satisfying $|\ell| \approx \chi$.

Using the sorter, the OAM content of the wavefunction after its interaction with the dipole was recorded and is shown in Fig. 3c. We find that the beam's OAM content is mostly distributed near $\ell = \pm 5$, thus implying that the dipole is defined by a $\chi$ value of $\sim 5$ rad. This value roughly translates to a magnetic dipole moment of $\mathcal{M} \approx 6.2 \times 10^9 \mu_B$, where $\mu_B$ is the Bohr magneton, and is in good agreement with our estimated value of the structure's saturated magnetic dipole moment of $6.7 \times 10^9 \mu_B$. The corresponding numerically simulated OAM spectrum based on these parameters is also included in Fig. 3c. These numerical results were obtained using our simplified model where we assume that $\chi(r)\sin\varphi \approx \chi(r_{max})\sin\varphi$ and are in good agreement with data based on the saturation field of the wire.

## Discussion

In comparison with other methods used to examine magnetic fields (c.f., refs 21,30,31), the OAM sorter proves its effectiveness by readily providing the beam's OAM spectrum. For an identical

number of detected electrons, this content is defined by an image showing 20 $|c_\ell|^2$ OAM coefficients yielding more information about a beam's phase than an image obtained using holographic methods. Moreover, such images do not allow a direct measurement of a sample's magnetic information. Instead, this quantity has to be extrapolated from the field in the dipole's proximity. In addition, the OAM sorter method does not rely on any phase wrapping or unwrapping methodology, thereby simplifying a magnetic field's analysis. Further developments could also allow the improvement of this device and the possibility to exploit it in atomic scale measurements or in conjunction with scanning electron probes. Given that the sorter only requires the phase masks in two distinct planes, then its performance will become more effective if absorptive elements like phase holograms were substituted by structured electrostatic fields[32].

Our sorter method also possesses the following prospective extensions. On the one hand, when using the sorter to analyse magnetic structures, the radial dependence of the phase added by such structures can be lowered by exposing them to a beam already carrying a known OAM value. This will cause the beam to have a maximal intensity at a certain radius $r_\ell$. Therefore, the majority of the beam's electrons that have interacted with the structure acquire a phase whose radial dependence will be attributed to $r_\ell$. Much like how we approximated the phase acquired by electrons to be predominantly defined by the truncation radius of our apparatus $r_{max}$, we could likewise assign additional importance to electrons attributed to a radius of $r_\ell$. Such an approximation could provide additional simplifications that are needed for matching the outcome of these magnetic measurements with theory. On the other hand, a minor modification to the sorter's schematics can be proposed to sort electron modes in a different mutually unbiased basis, namely the so-called angular basis[26]. Performing an additional set of measurements of an electron's phase in this basis could provide additional information, for example, regarding the phase of the

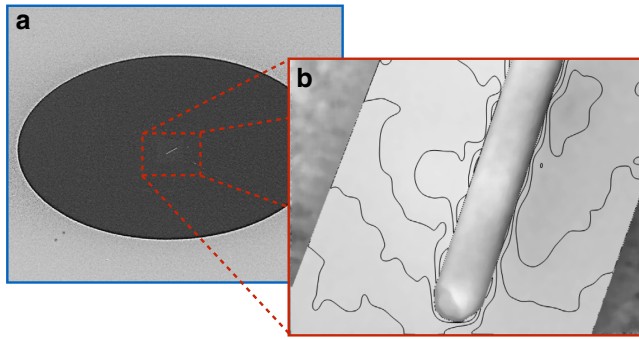

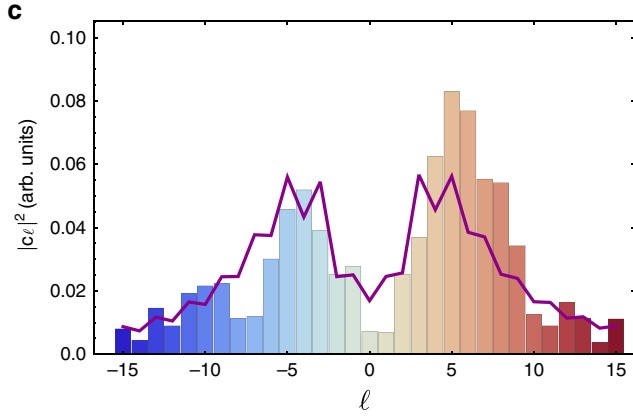

**Figure 3 | OAM spectrum of a beam affected by a magnetic dipole.**
(**a**) Scanning electron microscopy (SEM) image of the analysed magnetic bar configurated as a dipole. The bar is defined by dimensions of 100 nm (thickness) by 200 nm (width) by 2.8 μm (length). (**b**) Magnetic field lines of the magnetic bar measured using electron holography. (**c**) Expected (curve) and obtained (bars) OAM spectrum acquired by the electron beam upon interacting with the magnetic bar. The expected curve was calculated assuming that its magnetic field is saturated. Unlike the measurement performed in (**b**), the experimental data were obtained while the dipole was exposed to a field in the condenser plane of the electron microscope.

OAM expansion coefficients $c_\ell$[33], concerning the interaction of a magnetic field with electron beams.

**Data availability**. All relevant data are available from the authors on reasonable request.

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

## Acknowledgements

V.G. acknowledges the support of the Alexander von Humboldt Foundation. H.L., F.B., R.W.B. and E.K. acknowledge the support of the Canada Research Chairs (CRC) and Canada Excellence Research Chairs (CERC) Program. R.D.-B. is grateful to the European Research Council for funding under the European Union's Seventh Framework Programme (FP7/2007–2013)/ERC grant agreement number 320832.

## Author contributions

V.G., R.W.B, R.E.-D., M.J.P. and E.K. conceived the idea; M.J.P, V.G., M.P.J.L., and E.K. designed the hologram; F.V., G.C.G., P.-H.L., and E.M. fabricated the hologram; F.V., G.C.G., and S.F. fabricated the magnetic pillar; A.H.T., V.G., S.F. and R.B. performed the experiment; V.G., and A.H.T analysed the data; V.G., H.L., F.B., and E.K developed the application theory; H.L., F.B., and E.K. with the help from other authors wrote the manuscript. All authors discussed the results and contributed to the text of the manuscript.

## Additional information

**Competing interests:** The authors declare no competing financial interests.

**Publisher's note**: 

