## [Peer Review File · Nature Communications]

Reviewers' comments:

Reviewer #1 (Remarks to the Author):

Referee report on: Manuscript NCOMMS-16-26270

Authors: Ebrahim Karimi, Vincenzo Grillo, Amir Tavabi, Federico Venturi, Hugo Larocque, Roberto Balboni, Gian Carlo Gazzadi, Stefano Frabboni, Peng-Han Lu, Erfan Mafakheri, Frédéric Bouchard, Rafal Dunin-Borkowski, Robert Boyd, Martin Lavery, and Miles Padgett.

In this paper the authors had sought to implement a geometrical transformation to map mixed electron vortex modes into laterally positioned well separated modes. As an illustration they considered a superposition state with winding numbers -5 and $+5$ and expected that their design would ensure observation of two spatially separated spectral lines. The 'OAM-sorting' idea for electron vortex modes is apparently borrowed from the optical vortex context and was pioneered by the Glasgow optics group [reference 25].

Unfortunately, it seems that the design here is somewhat crude and clearly the technique is in need of further testing to ensure that it succeeds not just in simply generating the separate lines, but must also say something definite about the accuracy of the resolution of such lines, i.e. the ability to distinguish different lines as well as the purity of the sorting (referred to as cross-talk). These matters have not been given due consideration in this report. Perhaps the authors are aware that their masks are indeed rather poor, but nevertheless wished to put forward the idea for implementation in the electron vortex context.

In this referee's opinion, this work is interesting, albeit that the idea is not novel since one expects what can be done in the optics context to be, in principle, also possible in the electron vortex context. The devil is in the detail. I feel that the work is publishable in some form, but not in Nature Communications.

Specific comments on the manuscript are as follows:

1. Abstract: '...device measuring the azimuthal wavefunction..' is misleading and over-simplified even in a nature Communications abstract. The fact is what is meant is a measurement of a characteristic (namely the OAM) of the wavefunction.
2. Page 1, left column: reference 4 is cited as evidence of magnetic dichroism. In fact, that article says that no dichroism could be detected because of the symmetry between matrix elements for states with $-l$ and $+l$.
3. Page 3, left column '...However, due to fabrication and alignment imperfections in our apparatus, including the holograms generating the OAM carrying electrons, we observe higher values of cross-talk between OAM components'. Here the authors admit that FIG.1 is not evidence of a pure state $l=5$, but a distribution around that value. No mention of the resolution of the different channels at this stage.
4. Page 3, right column, equation (1) needs further details of the derivation and references.
5. Page 3, right column: 'However, given that the resulting expansion terms carry a quantized azimuthal phase defined by \dots ', then it follows that these components also carry OAM values of...'. By default, the expansion coefficients c correspond to the weight of each OAM component of a beam having been affected by the magnetic dipole.'. Not clear. This statement and the rest of the para need careful re-writing to clarify how the magnetic moment can be ascertained via the Jacobi-Anger formalism.
6. Page 4, left column: 'The use of this model is justified by the truncation of the beam after passing

through the first of the sorter's holograms. Such truncations result in lensing effects which will consequently displace electrons that were originally positioned at the hologram's cutoff radius r_{\max} '. This is the source of broadening effect which requires careful consideration by the authors in the context of their experimental findings.

Reviewer #2 (Remarks to the Author):

The authors propose an approach to measuring the OAM spectra of electron beam. The principle is discussed briefly, and verified by affecting electrons with a micron-scale magnetic dipole. The idea behind this work is meaningful and fits well in the research horizons of the scientific community. I recommend this manuscript for publication in Nature Communications, after doing modifications mentioned below.

1. The manuscript is not structured according to the format of NC, which need to be improved.
2. The OAM mode sorter here is widely employed for the detection of OAM spectra of photons, where a lens is placed between sorter and corrector, and the corrector is placed in the Fourier plane of the lens (Phys. Rev. Lett. 105,153601(2010)). However, in this manuscript, I can't see any lens in such locations. Some explanation should be given.
3. A paper published in NC must be repeatable. Thus a detail experimental setup should be presented, including the element, model of instruments and so on. Such statement can be shown in the part of methods, or supplementary materials.
4. The paragraph below Fig. 2, "However, due to fabrication and alignment imperfections in our apparatus, including the holograms generating the OAM carrying electrons, we observe higher values of cross-talk between OAM components." How to overcome such cross-talk?

Reviewer #3 (Remarks to the Author):

This is a nice work, which describes an OAM sorter for electron vortex beams. The idea originates from the analogous optical devices described in Refs. [25,26]. Namely, a universal phase element breaking the cylindrical symmetry of the OAM eigenstates transforms their azimuthal quantum numbers l into l -dependent transverse displacements. Such "OAM spectroscopy" has been proved to be useful for various optical applications.

In the work under consideration, the authors demonstrate its efficiency for the electron OAM probing of magnetic materials, which is one of the main potential application areas for electron vortex beams. The manuscript is clearly written, and the experimental results are convincing. Therefore, I recommend its publication in Nature Communications.

A minor remark: There are several earlier works which suggested measurements of electron OAM using various interferometric methods. The authors briefly discuss these (Ref. [20,22]) in the Introduction and argue about the advantages of the spatial OAM sorting. I think one paper is missing here and should be also discussed:

L. Clark et al., "Quantitative measurement of orbital angular momentum in electron microscopy," Phys. Rev. A 89, 053818 (2014).

We would like to thank all three referees for providing us with their concerns and suggestions which have significantly helped us with improving the quality and depth of our manuscript. A list of the comments given to us in our previous correspondence is provided below along with our responses to them and the way we accordingly modified our manuscript.

Referee #1: *In this paper the authors had sought to implement a geometrical transformation to map mixed electron vortex modes into laterally positioned well separated modes. As an illustration they considered a superposition state with winding numbers -5 and +5 and expected that their design would ensure observation of two spatially separated spectral lines. The 'OAM-sorting' idea for electron vortex modes is apparently borrowed from the optical vortex context and was pioneered by the Glasgow optics group [reference 25].*

Unfortunately, it seems that the design here is somewhat crude and clearly the technique is in need of further testing to ensure that it succeeds not just in simply generating the separate lines, but must also say something definite about the accuracy of the resolution of such lines, i.e. the ability to distinguish different lines as well as the purity of the sorting (referred to as cross-talk). These matters have not been given due consideration in this report. Perhaps the authors are aware that their masks are indeed rather poor, but nevertheless wished to put forward the idea for implementation in the electron vortex context.

In this referee's opinion, this work is interesting, albeit that the idea is not novel since one expects what can be done in the optics context to be, in principle, also possible in the electron vortex context.

Reply: Whereas it is in principle true that most things in optics can be reproduced with electron optics, the practical realization for electrons is extremely complicated. Namely, only one work so far involved the use of 2 holograms. In our work, we use three holograms, a custom electron lens setup, and a full-scale simulation of the electronic column. But this is not the most relevant factor here.

The most relevant factor of our work is that a well-known technique in optics (the so called "Glasgow" sorter) discloses completely new opportunities in electron microscopy of magnetic materials. In particular, we emphasized this aspect of our work when we used our electron sorter to analyze the magnetic structure of a magnetic dipole. We have also provided discussions concerning the advantages of this method over current methods relying primarily on holography. In fact it is really worthwhile stressing that in this new acquisition method, the extracted magnetic information has been compressed from a series of (projective) images to a single spectrum. We are therefore going in the direction of measuring a quantity in microscopy with potentially very few electrons.

Referee #1: *The devil is in the detail. I feel that the work is publishable in some form, but not in Nature Communications.*

Reply: We took great care of details with an experiment that lasted 3 years and required the top nanofabrication techniques and the top electron microscopes. In fact, only the *FEI Titan HOLO* (a quite rare instrument in his own right) is one of the instruments with the extra lens necessary for this experiment.

Referee #1: *Specific comments on the manuscript are as follows:*

1. Abstract: ‘... device measuring the azimuthal wavefunction...’ is misleading and over-simplified even in a nature Communications abstract. The fact is what is meant is a measurement of a characteristic (namely the OAM) of the wavefunction.

Reply: To address this over-simplification, we have changed “measuring the azimuthal wavefunction” to “measuring an electron's OAM components”.

Referee #1: *2. Page 1, left column: reference 4 is cited as evidence of magnetic dichroism. In fact, that article says that no dichroism could be detected because of the symmetry between matrix elements for states with $-l$ and $+l$.*

Reply: We have removed reference 4 from our list of cited works.

Referee #1: *3. Page 3, left column ‘...However, due to fabrication and alignment imperfections in our apparatus, including the holograms generating the OAM carrying electrons, we observe higher values of cross-talk between OAM components’. Here the authors admit that FIG.1 is not evidence of a pure state $l=5$, but a distribution around that value. No mention of the resolution of the different channels at this stage.*

Reply: As highlighted in Supplementary Note 3, there are clear selection rules even for imperfect holograms. For example, the $+5 -5$ superposition beam should not generate any component for $l=4$. Any intensity on this channel is due to the cross talk. This is therefore an absolute measurement of the resolution of the apparatus. Namely, the source of cross-talk in the device specifically consists of an effect inherent to the implemented geometric transformation used to sort the beam's OAM content. Thereby, this effect (the impression of having a distribution around a specific OAM value in the beam's final profile) is also observed in a theoretical treatment of the process where ideal phase elements are used to perform the required conformal mapping. The aim of our comments regarding fabrication imperfections is not to identify them as the source of cross-talk in our experiment, but rather as an additional factor that increases the cross-talk inherent to the process. So far, the only way to overcome the cross-talk inherent to this transformation consists of using a so called “fan-out” configuration for the sorter as employed in optical sorters [26]. However, this methodological adaptation requires holograms with a resolution high enough to make the implementation of such a fan-out electron sorter a work of its own. Though holograms with a higher quality could allow the implementation of this fan-out configuration, another way to implement it would be to include two additional confocal holograms to the current configuration. Although, this would add more electron losses to the sorting process and is not configurable in our electron microscope.

Referee #1: *4. Page 3, right column, equation (1) needs further details of the derivation and references.*

Reply: We were not sure whether this comment addressed equation (1) on page 2 or equation (2) on page 3. Therefore, we provided additional details in our revised manuscript regarding both. For equation (1), we have rewritten the passage leading to the equation as:

“The coordinates (u,v) are formally known as log-polar coordinates and can be related to the Cartesian coordinates (x,y) via the transformations $x=\exp(u)\cos(v)$ and $y=\exp(u)\sin(v)$ or equivalently by $u=\ln(\sqrt{x^2+y^2})$ and $v=\arctan(y/x)$ [27]. A Cartesian coordinate $x+iy$ in the complex plane can be mapped to its corresponding log-polar coordinate $u+iv$ with the conformal mapping $f(z)=\ln(z)$. Using a scalable version of this mapping, $\Phi(z)=a\ln(z/b)$, an OAM-carrying beam's transverse wavefunction $\psi_f(r,\varphi)=f(r)\exp(il\varphi)$ can be mapped to the following wavefunction.”

Note that we have added reference [27] which describes the mapping in the optical scenario. As for further details involving equation (2), we have added an outline of its derivation in Supplementary Note 5.

Referee #1: 5. Page 3, right column: ‘However, given that the resulting expansion terms carry a quantized azimuthal phase defined by ...’, then it follows that these components also carry OAM values of...’. By default, the expansion coefficients c correspond to the weight of each OAM component of a beam having been affected by the magnetic dipole.’ Not clear. This statement and the rest of the para need careful re-writing to clarify how the magnetic moment can be ascertained via the Jacobi-Anger formalism.

Reply: Based on our understanding, this comment addresses a lack of progression between the dipole structure and the weights of the OAM components that it imparts on the electron beam. We believe that the added derivation of the $g(r,\varphi)$ function in Supplementary Note 5 should clarify this link. The passage highlighted by the referee appears to be clear as it is since it simply describes a Fourier expansion of the $g(r,\varphi)$ function (equation (2)) and how each of its components carry OAM. However, to further clarify how the Jacobi-Anger expansion is used to obtain the weight of each OAM component, we have added the expansion right after the formal Fourier expansion in this passage for readers to be able to see the identical form of both equations.

Referee #1: 6. Page 4, left column: ‘The use of this model is justified by the truncation of the beam after passing through the first of the sorter’s holograms. Such truncations result in lensing effects which will consequently displace electrons that were originally positioned at the hologram’s cutoff radius r_{max} ’. This is the source of broadening effect which requires careful consideration by the authors in the context of their experimental findings.

Reply: Though this truncation does cause broadening in the beam, it does not change its OAM content and thus does not influence the final output of our device. The “broadening” in our reported OAM spectra simply consists of the aforementioned inherent cross-talk of our device. Furthermore, the presence of this in-built aperture does not significantly affect the beam’s final spectrum. Indeed, this aperture is located near the sorter’s imaging focus and therefore does not introduce a considerable amount of broadening to the electron beam. This was theoretically verified via numerical simulations.

Referee #2: *The authors propose an approach to measuring the OAM spectra of electron beam. The principle is discussed briefly, and verified by affecting electrons with a micron-scale magnetic dipole. The idea behind this work is meaningful and fits well in the research horizons of the scientific community. I recommend this manuscript for publication in Nature Communications, after doing modifications mentioned below.*

1. The manuscript is not structured according to the format of NC, which need to be improved.

Reply: We would like to thank the referee for pointing that out. We have modified our manuscript to the Nature Communications format.

Referee #2: *2. The OAM mode sorter here is widely employed for the detection of OAM spectra of photons, where a lens is placed between sorter and corrector, and the corrector is placed in the Fourier plane of the lens (Phys. Rev. Lett. 105,153601(2010)). However, in this manuscript, I can't see any lens in such locations. Some explanation should be given.*

Reply: The lenses that we used in our implementation consist of our electron microscope's in-built lenses. We have added a detailed experimental setup in the supplementary information (Supplementary Figure 1) showing the lens configuration of the microscope along with the following passage in the main text.

"We adopt a similar diffractive approach to develop our sorter using two electron phase holograms, as displayed in Fig. [1], and where the lensing effects required to perform the mode transformation are configured using our electron microscope's lenses (see Supplementary Note 1 and Supplementary Figure 1 for more details)."

Referee #2: *3. A paper published in NC must be repeatable. Thus a detail experimental setup should be presented, including the element, model of instruments and so on. Such statement can be shown in the part of methods, or supplementary information.*

Reply: We have added a part in the supplementary information (Supplementary Figure 1 and Supplementary Note 1) providing our experimental setup where the lenses, holograms, and the electron beam's evolution through the microscope are depicted.

Referee #2: *4. The paragraph below Fig. 2, "However, due to fabrication and alignment imperfections in our apparatus, including the holograms generating the OAM carrying electrons, we observe higher values of cross-talk between OAM components." How to overcome such cross-talk?*

Reply: The cross-talk can be overcome in part by improving the quality of the employed holograms. However, secondary methods can also be used to lower this limitation such as adopting a "fan-out"

configuration of the device as commonly used in our sorter's optical counterpart. Such a configuration is presented in [26]. Moreover, the aberrations originating from the strong XL lens' excitation could also be lowered in different experimental configurations which could lower the observed cross-talk. To provide the reader with this insight, we have added the following passage to our manuscript after reporting our cross-talk values:

“Such experimental limitations could be overcome by adopting a fan-out configuration as employed in optical sorters [25,26] and by improving the quality of the sorter and the corrector holograms via alternative fabrication methods. Cross-talk can also be further reduced by an improved control over the electron beam's aberrations.”

Referee #3: *This is a nice work, which describes an OAM sorter for electron vortex beams. The idea originates from the analogous optical devices described in Refs. [25,26]. Namely, a universal phase element breaking the cylindrical symmetry of the OAM eigenstates transforms their azimuthal quantum numbers l into l -dependent transverse displacements. Such “OAM spectroscopy” has been proved to be useful for various optical applications. In the work under consideration, the authors demonstrate its efficiency for the electron OAM probing of magnetic materials, which is one of the main potential application areas for electron vortex beams. The manuscript is clearly written, and the experimental results are convincing. Therefore, I recommend its publication in Nature Communications.*

A minor remark: There are several earlier works which suggested measurements of electron OAM using various interferometric methods. The authors briefly discuss these (Ref. [20,22]) in the Introduction and argue about the advantages of the spatial OAM sorting. I think one paper is missing here and should be also discussed:

L. Clark et al., “Quantitative measurement of orbital angular momentum in electron microscopy,” Phys. Rev. A 89, 053818 (2014).

Reply: We have added this reference to our list of cited works (reference 24). We cite it in the passage where we discuss interferometric methods. We believe that further discussions do not have to be brought to the passage since it already discusses the limitations of interferometric methods, i.e., the stability of the interferometer and the coherence of the electrons.

REVIEWERS' COMMENTS:

Reviewer #1 (Remarks to the Author):

2nd round: referee report on: NCOMMS-16-26270

Authors: Ebrahim Karimi et al

The authors have made significant changes to the original manuscript in response to my first report and put forward robust arguments in defense of my points of reservations for publication of the work in Nature Communications.

In particular, the main text as well as the supplementary material now contain useful extensions which have clarified some of the previously unclear parts of the manuscripts.

The authors also make sufficient emphasis on the fact that the work needs further refinement to make the technique accessible in the future if it is to be useful, as they say, in probing magnetic materials.

Reviewer #2 (Remarks to the Author):

The manuscript was revised according to my comments. It can be accepted for publication.

Reviewer #3 (Remarks to the Author):

The authors successfully addressed the issues raised by the Referees, and the paper can be published in Nature Communications in the present form.

We would like to thank all three referees for providing us with their concerns and suggestions which have significantly helped us with improving the quality and depth of our manuscript. A list of the comments given to us in our previous correspondence is provided below along with our responses to them and the way we accordingly modified our manuscript.

Referee #1: *2nd round: referee report on: NCOMMS-16-26270*

Authors: Ebrahim Karimi et al

The authors have made significant changes to the original manuscript in response to my first report and put forward robust arguments in defense of my points of reservations for publication of the work in Nature Communications.

In particular, the main text as well as the supplementary material now contain useful extensions which have clarified some of the previously unclear parts of the manuscripts.

The authors also make sufficient emphasis on the fact that the work needs further refinement to make the technique accessible in the future if it is to be useful, as they say, in probing magnetic materials.

Reply: We thank the referee for his comments.

Referee #2: *The manuscript was revised according to my comments. It can be accepted for publication.*

Reply: We thank the referee for his comments.

Referee #3: *The authors successfully addressed the issues raised by the Referees, and the paper can be published in Nature Communications in the present form.*

Reply: We thank the referee for his comments.